# Efficient metal halide perovskite light-emitting diodes with significantly improved light extraction on nanophotonic substrates

Qianpeng Zhang[1,2], Mohammad Mahdi Tavakoli[1], Leilei Gu[1,2], Daquan Zhang[1], Lei Tang[1], Yuan Gao[1], Ji Guo[1], Yuanjing Lin[1], Siu-Fung Leung[1,3], Swapnadeep Poddar[1], Yu Fu[1] & Zhiyong Fan[1,2]

Metal halide perovskite has emerged as a promising material for light-emitting diodes. In the past, the performance of devices has been improved mainly by optimizing the active and charge injection layers. However, the large refractive index difference among different materials limits the overall light extraction. Herein, we fabricate efficient methylammonium lead bromide light-emitting diodes on nanophotonic substrates with an optimal device external quantum efficiency of 17.5% which is around twice of the record for the planar device based on this material system. Furthermore, optical modelling shows that a high light extraction efficiency of 73.6% can be achieved as a result of a two-step light extraction process involving nanodome light couplers and nanowire optical antennas on the nanophotonic substrate. These results suggest that utilization of nanophotonic structures can be an effective approach to achieve high performance perovskite light-emitting diodes.

[1] Department of Electronic and Computer Engineering, The Hong Kong University of Science and Technology, Clear Water Bay, Kowloon, Hong Kong SAR, China. [2] HKUST-Shenzhen Research Institute, No. 9 Yuexing first RD, South Area, Hi-tech Park, Nanshan, Shenzhen 518057, China. [3] Present address: Department of Computer, Electrical and Mathematical Sciences and Engineering, King Abdullah University of Science and Technology, Thuwal 23955-6900, Saudi Arabia. Correspondence and requests for materials should be addressed to Z.F. (email: eezfan@ust.hk)

Metal halide perovskites have triggered tremendous research interest due to their favorable optoelectronic properties such as wide-range tunable bandgap, balanced carrier mobility, and long carrier diffusion lengths, high luminescence efficiency, etc[1–3]. In the past few years, intensive research on optimization of perovskite film composition, morphology, and interface engineering has dramatically improved the perovskite solar cell efficiency from 3.8% to over 22%[4]. Besides photovoltaics, hybrid perovskites also promise a bright future for light-emitting diodes (LEDs) owing to their desirable properties such as high color purity, high luminance yield, low non-radiative recombination rates, and suppressed defects formation[5,6].

Up until now, significant progress has been made on improving perovskite LED performance by material engineering including optimizing perovskite grain size, exploring materials for charge injection, utilizing conducting polymers mixed with perovskite, and using quantum confinements, etc[7–13]. Despite the considerable improvement has been made, it is worth mentioning that the overall external quantum efficiency (EQE) of $CH_3NH_3PbBr_3$ (Br-Pero) LEDs is still low in most cases. One of the important reasons accounts for this is the low light extraction efficiency which is caused by large refractive index contrast between the active layer and charge injection layers. Specifically, according to Snell's Law, only the light within a narrow escape cone $\theta < a\sin(n_{ambient}/n_{active})$ can be extracted vertically from a LED device, where $n_{ambient}$ is refractive index of the ambient medium and $n_{active}$ is refractive index of the active layer. Due to the high refractive index of perovskite materials and low refractive indexes of injection layers, a large portion of the generated light is confined in the perovskite material and propagates in the lateral direction, which compromises light extraction efficiency. For example, for a widely adopted material system ITO/PEDOT: PSS/Br-Pero/F8/Ca/Ag, the refractive indexes of Br-Pero, PEDOT: PSS and F8 at 550 nm wavelength are 2.2, 1.5, and 1.7, respectively[14,15]. Such a large refractive index difference between Br-Pero and PEDOT: PSS/ F8 can easily cause total internal reflection either from PEDOT: PSS side or from F8 side. Therefore, the Br-Pero layer is undesirably functioning as a planar waveguide providing a lateral light pathway. Note that for typical organic LEDs (OLEDs), the refractive index difference between the active materials and injection layer is typically not this large, thus light extraction efficiency is not a significant performance limiting factor. However, for an organic-inorganic hybrid material system, the large refractive index difference is detrimental. It has been reported that rationally designed nanophotonic structures on top of optoelectronic devices can help with light coupling. For instance, nanostructures on solar cell devices can reduce front surface reflection and improve device power conversion efficiency[16–20]. In these cases, the nanostructures can improve refractive index matching between the solar cell active materials and air, thus significantly increase light in-coupling efficiency. Recently, there are some works devoted to address the light extraction problems with microlens or hollow fiber structures and achieved a few times enhancement for OLED performance[21,22]. However, this approach has not been adopted for metal halide Pero LEDs to our best knowledge. Herein, we have fabricated perovskite thin film LEDs on the barrier layer side of anodic alumina membranes (AAMs) which work as three-dimensional (3-D) nanophotonic substrates. And a nanophotonic substrate consists of a layer of nanodome (ND) array light coupler and a layer of nanowire (NW) array optical antennas.

In our work, $CH_3NH_3PbBr_3$ (Br-Pero) is chosen as the model material system to demonstrate the device concept with emission wavelength at the center of luminosity function. We have fabricated AAMs with different nanostructure geometries to accommodate Br-Pero LED devices and discovered that a two-time

enhancement of EQE, from 8.19% for a typical planar control device to 17.5% for the optimized nanostructured device, could be achieved. To our best knowledge, this EQE value is the highest for Br-Pero LEDs and it is also close to twice of current EQE record for this material system[23]. Moreover, optical modeling with Finite-Difference Time-Domain (FDTD) method has been performed to gain more in-depth understanding on the role of the nanophotonic structure for Pero LED performance enhancement. And the EQE enhancement in experiment is supported by optical modeling showing that the NDs serve as light couplers to focus light into NW array photonic crystals which then work as optical antennas to convert confined energy in guided modes to leaky modes (scattering resonance). When using the optimal nanophotonic design, a high light extraction efficiency of 73.6% can be achieved. It is worth mentioning that the unique nanophotonic LED substrate demonstrated in this work can be also used for other perovskite material systems to achieve high performance in the future.

## Results

**Device structure**. Figure 1a schematically shows the device structure of the Pero LED. It can be seen that the device consists of 7 layers of materials in addition to a supporting substrate and an adhesion layer, i.e., AAM(TiO₂ NWs)/ ITO/ PEDOT: PSS/Br-Pero/F8/Ca/Ag (details of material properties are in Table 1). In this device structure, the AAM(TiO₂ NWs) layer is a photon management structure consisting of a hexagonal array of NDs as the barrier layer and a layer of TiO₂ NW array embedded in AAM. Figure 1b shows the cross-sectional scanning electron microscopy (SEM) image of an AAM. Different materials were then sequentially deposited on the AAM NDs barrier layer to form a LED device and the detailed process can be found in the Methods section. The AAM shown here has a thickness of about 3.5 μm with a uniform pore diameter of 120 nm and pitch of 500 nm. More SEM images of the AAMs can be found in Supplementary Figure 1. Supplementary Figure 2a shows the energy band alignment of the device in which F8 layer and PEDOT: PSS layer serve as the electron and hole injection layers, and Br-Pero serve as the light emission layer, respectively. Figure 1c shows the cross-sectional SEM image of a completed nanostructured device on an AAM substrate with 500 nm pitch (P500). SEM images of thin film (TF) control device and P1000 AAM device can be found in Supplementary Figure 2b and 2c. The detailed device fabrication process can be found in the Methods section. It can be seen that each layer of material can be clearly resolved. Detailed thickness and fabrication conditions can be found in Methods. The X-ray diffraction (XRD) pattern, UV-Visible spectrum, photoluminescence (PL) and time resolved photoluminescence (TRPL) have been obtained and shown in Supplementary Figure 3, and related discussion is in Supplementary Note 1. These results confirm good quality of the prepared Br-Pero thin film[24].

**Device performance**. After device fabrication and material characterization, the device performance measurement was then carried out. The current density–voltage (J–V) curves for devices with different structures are shown in Figure 2a. Thin film device showed higher current density than nanostructured device. This can be attributed to the higher sheet resistance of our sputtered ITO on nanostructure compared with the planar ITO coated substrate. (Supplementary Figure 4.) It is crucial that utilization of nanostructures does not increase the current density, as too high current density may hurt the internal quantum efficiency (IQE) by increasing the current leakage and non-radiative recombination[25]. It is worth noting that the photonic crystal structure and the device active layer are preferred to be separated so that the

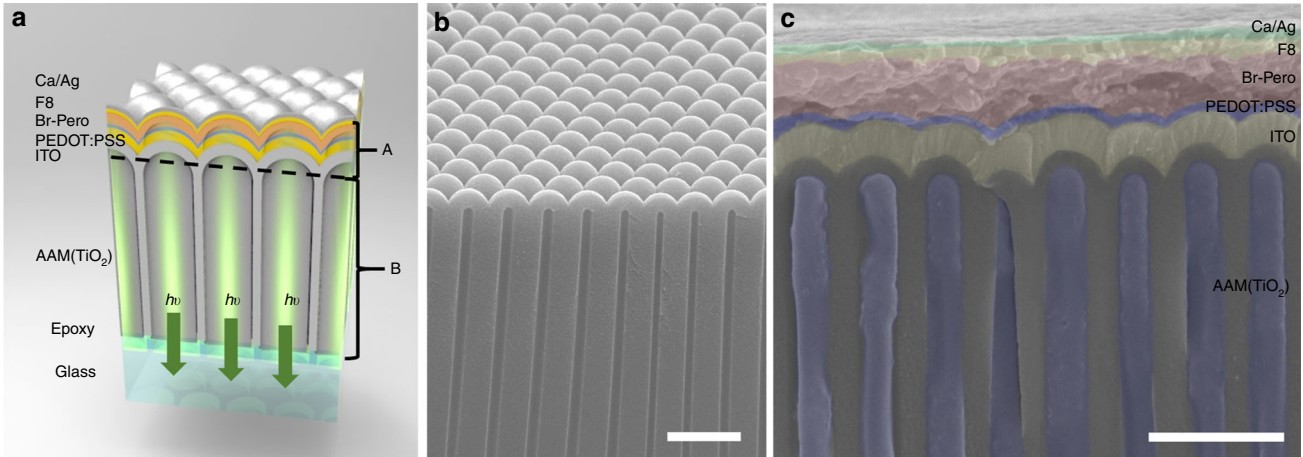

**Fig. 1** Device on nanophotonic substrate. **a** Device schematic. The materials from top to bottom are: Ca/Ag electrode, F8, CH₃NH₃PbBr₃ (Br-Pero), PEDOT: PSS, ITO, and anodic alumina membrane (AAM). AAM channels are filled with TiO₂. **b** SEM image of the barrier side of the free-standing AAM film with nanodomes structures. **c** Cross-sectional SEM image of a P500 AAM device. Scale bars in **b** and **c** are 1 μm

| Table 1 Material details for the Br-Pero nanophotonic LED device structure | | | | | | | |
|---|---|---|---|---|---|---|---|
| **Material** | **Function** | **Work function (eV)** | **$E_C$ (eV)** | **$E_V$ (eV)** | **HOMO (eV)** | **LUMO (eV)** | **Refractive index** |
| Anodic alumina (AAM) | Photon management | | | | | | 1.7 |
| Titanium oxide (TiO₂) | Photon management | | | | | | 2.6 |
| Indium-doped tin oxide (ITO) | Anode | 4.7 | | | | | |
| Poly(3,4-ethylenedioxythiophene): poly (styrenesulfonate) (PEDOT: PSS) | Hole injection layer (HIL) | | | | 5.2 | | 1.5 |
| BA: CH₃NH₃PbBr₃ (Br-Pero) | Active layer | | 3.4 | 5.7 | | | 2.2 |
| Poly(9,9'-dioctyl-fluorene) (F8) | Electron injection layer (EIL) | | | | | 2.9 | 1.7 |
| Calcium/ silver (Ca/ Ag) | Cathode | 3 | | | | | |

electrical performance would not be compromised too much by introducing the structure. The luminance-voltage curves of Pero LEDs with different structures are also shown in Figure 2a. Intriguingly, the luminance can be greatly enhanced by optimizing the geometry of nanostructure on substrate. Particularly, the luminance of planar, P500, P1000 devices under 5 V bias are 26,106 cd m⁻², 12,874 cd m⁻² and 48,668 cd m⁻², respectively. Compared to the planar control device, the luminance for P500 device was much reduced, however, the luminance for P1000 device was increased by 1.86 times as compared with the planar control device. The inset of Figure 2a are the optical photo of different devices lighted up with 3.5 V driving voltage. It can be seen that the P1000 device has the highest brightness and the P500 device has the lowest brightness with the same camera settings, which means a geometrical design optimization is required in order to obtain an enhanced performance. Moreover, the EQE versus voltage curves of different devices are illustrated in Figure 2b. The EQE increases from 0 to 4 V, reaching the maximum value at around 4 V and starts to show roll-off in 4 to 5 V voltage range. The EQE roll-off is a common and major issue for most LEDs, especially for perovskite LEDs. For perovskite LED, this roll-off is mainly caused by non-radiative Auger recombination with high injection current density and it can be addressed by methods such as using multiple-quantum-wells[26,27]. Beyond 5 V, the devices start to fail due to Joule heating[7]. The maximum EQE is 8.19% for the planar control device at 4.4 V which is comparable to the best-reported value of 9.3% for this particular material system[23]. (Refer to Supplementary Table 1.)

Meanwhile, the EQE is only 3.0% for P500 device at 4.4 V, however it is 17.5% for P1000 device at 4.0 V. Therefore, the peak EQE for P1000 device was increased to twice of that of the planar device. (The histograms of the EQE distribution are shown in Supplementary Figure 5.) To our best knowledge, this 17.5% EQE is so far the highest record in literature for Br-Pero LEDs, it is close to twice of the reported record for this material system[7]. And this set of results clearly demonstrates the critical role of rational nanophotonic design in determining device performance.

To further examine the EQE improvement, the EQEs of nanostructured devices, namely P500 and P1000 devices, were normalized to that of the planar control device at the same voltage and the enhancement factor (EF, defined as the EQE ratio of the nanophotonic devices over the planar device) vs voltages are plotted in Figure 2c. Obviously, the EQE EF for P1000 device firstly increases and reaches the maximum at 3.6 V then decreases. However, the EQE EF for P500 device is only around 0.5 in most part of 2–5 V voltage range. This means the P500 structure is not favorable for light outcoupling. The maximum EQE EF is 2.94 at 3.6 V for P1000 device. To understand this interesting observation, light extraction mechanism in different structures must be explored with the assistance of optical simulations in this work which will be discussed in the following paragraphs.

**Optical simulation.** Our previous work has shown that FDTD simulation can clearly reveal the light propagation and absorption behaviors in nanostructures[16,28,29]. To gain more in-depth

understanding on the role of the nanophotonic structure for Pero LED performance enhancement, optical simulations with FDTD method were conducted to calculate the light extraction coefficient (EC) for devices with different geometries. Note that the light extraction efficiency is EC × 100%. The simulation structure can be found in Supplementary Figure 6a. Meanwhile, for the sake of comparison, the planar device structure was also modeled (Supplementary Figure 6b). The details of the simulation process can be found in the Methods section. And the result is shown in Figure 3a as a two-dimensional (2-D) color contour map with x-axis of the map as the pitch of the AAM nanopore/$TiO_2$ NWs array and y-axis as the diameter/ pitch (DP) ratio, namely, pore diameter divided by pitch. Note that the diameter of the AAM NDs is determined by the AAM pitch, as can be seen in Supplementary Figure 7 and 8. And the color depth represents EC. EC = 1 indicates that all light can be extracted from the device, and EC = 0 means that no light can escape from the structure. From Figure 3a, it can be found that the best light extraction condition corresponds to the structure with 1000 nm pitch. When the pitch is less than 500 nm, the light extraction is surprisingly low. Moreover, when the pitch is 1000 nm, the best light extraction (EC = 0.736) comes from the structure with the aspect ratio of 0.4. Additionally, this optimized geometry is a positive photonic crystal with fill fraction of 0.145 and effective refractive index of 1.86. The detailed calculation process can be found in Supplementary Note 2. However, under the same simulation conditions, the planar control device only showed an EC of 0.1, which means only 10% emitted photons can be extracted. Moreover, we made P1000 AAM devices with different DP ratios, and results are shown in Supplementary Figure 9. The device performance follows the same trend as our simulation results.

The above simulation results show that the EC of the optimized device structure is 7.36 times of that of a planar device. The modeled ECs and the EC- enhancement factors (EC-EFs), defined as the ratio of ECs of nanostructured device over the planar device, are shown in Table 2. However as shown in Figure 2b, in experiment the EQE EF is only 2.14. In order to explain the difference of EF between simulation and experiment, we provide the performance of our 3 generations of devices in the course of material optimization, as shown in Table 3. For the 1st generation device, pure $CH_3NH_3PbBr_3$ without any material engineering process was used. Both planar perovskite TF and perovskite on P1000 have PLQYs around 5%, and an EQE EF of 5.06 was achieved, which is closer to the 7.36 times enhancement from simulation results. For the 2nd generation device, we optimized the perovskite layer by crystal pining method. Then the planar TF perovskite PLQY was improved to 40% however the perovskite on P1000 substrate was only improved to 25%. In this case, the nanophotonic device EQE is only 3 times of the planar TF device. Furthermore, in the 3rd generation device, PLQY of the planar TF perovskite has been improved to 85% however the film on the nanophotonic structure was only improved to 35%. In this case, the TF device baseline was enhanced to 8.19%, which is close to 9.3% record of this particular material[7]. And the nanophotonic device EQE was improved to 2.14 times reaching up to 17.5%. Therefore, the EQE enhancement factor discrepancy is mainly from the different PLQY improvement factor which indicates different material quality improvement progress. And we attribute the reason accounts for this to different substrate surface morphology for planar substrate and the nanophotonic substrate. Namely, the optimal planar TF process conditions not necessarily yield the TFs with ideal quality on nanostructures. Therefore, the PLQY of perovskite on nanostructures was only improved from 4% up to 35%. This also suggests that perovskite process conditions on nanostructures need further optimization and with further such optimization, device performance can be substantially improved in the future. Moreover, the ITO transmittance on AAMs is found not as good as the commercial ITO glass that we used. As shown in Supplementary Figure 4, the ITO glass shows 80% transmittance at 550 nm while our sputtered ITO on AAM substrate shows only 50% transmittance. This is because of our nanophotonic structure has large surface area, in order to achieve a good electrical conductance, we reduced the $O_2$ concentration during ITO sputtering which compromises the transmittance. And this ITO transmittance loss also caused the EQE enhancement loss with respect to the modeling result.

With FDTD simulation, we have discovered that $TiO_2$ NWs filled in AAM nanopores play an important role in device performance enhancement. Specifically, we have replaced $TiO_2$ with air in simulation and the EC results are shown in Figure 3b. The EC follows similar trend to that in Figure 3a, nevertheless the highest EC is only 0.359 which is also obtained from 1000 nm pitch and 0.4 DP ratio. Although the EC of the device with air channels is 3 times of that of the planar device, it is much lower than that from device with $TiO_2$ NWs. A simple explanation is that the core index needs to be higher than that of $Al_2O_3$ (which is 1.7) in order to form waveguide to support guided modes coupled from perovskite layers. (Detailed mode analysis can be found in Supplementary Figure 10–13 and Supplementary Note 3.) Note that, theoretically any materials with a refractive index of 2.6 can replace the $TiO_2$ here. To verify the core refractive index effect, different core refractive indexes were simulated, and the results are in Supplementary Figure 14.

**Light extraction mechanisms**. In order to further acquire a clearer picture of the functions of the nanophotonic substrate, we divided it into two parts. One part (part A shown in Figure 1a) is the light coupling structure based on the NDs alumina barrier layer, and the other part (part B) is the photonic crystal optical antennas structure formed by hexagonal nanopore arrays filled up with $TiO_2$ NWs. As for part A, the NDs light couplers help to focus the light from the active material to the $TiO_2$ NWs. To visualize the light coupling effect, the dynamic process of light propagation in different structures was also simulated. The E field change with time elapses for different structures are summarized in Figure 4 and Supplementary Figure 15. The animation of light propagation in nanostructures is also available in Supplementary Movie 1–5. And it can be clearly seen that, with the help of NDs, light is coupled into the $TiO_2$ NWs a few femtoseconds after it is generated in the perovskite layer, however, the light is adversely confined inside the perovskite layer for planar device case. Supplementary Figure 16d, e and f clearly show the light focusing point of P500, P1000, and P1500 NDs without the NWs photonic crystal part. And light extraction efficiency for P500, P1000 and P1500 device are 20%, 46%, and 43.5%, respectively. As for part B, the nanopores are filled with high refractive index material ($n_{TiO2} = 2.6, n_{Alumina} = 1.7$) to form a positive photonic crystal and with a proper geometry it can work as optical antennas. Note that the P500, P1000, and P1500 device with a combination of both NDs light coupler and photonic crystal optical antennas show light extraction of 5.2%, 73.6%, and 26.9%, respectively. Only P1000 device has significantly enhanced light extraction compared to the case with only NDs. This is a strong evidence that the photonic crystal optical antennas play a key role in the light extraction process and the optimized geometry is needed in order to have the best enhancement. More discussion can be found in Supplementary Note 4.

To obtain further understanding of the light extraction mechanisms for different structures, cross-sectional Electric field square ($E^2$) intensities in the frequency domain are shown in

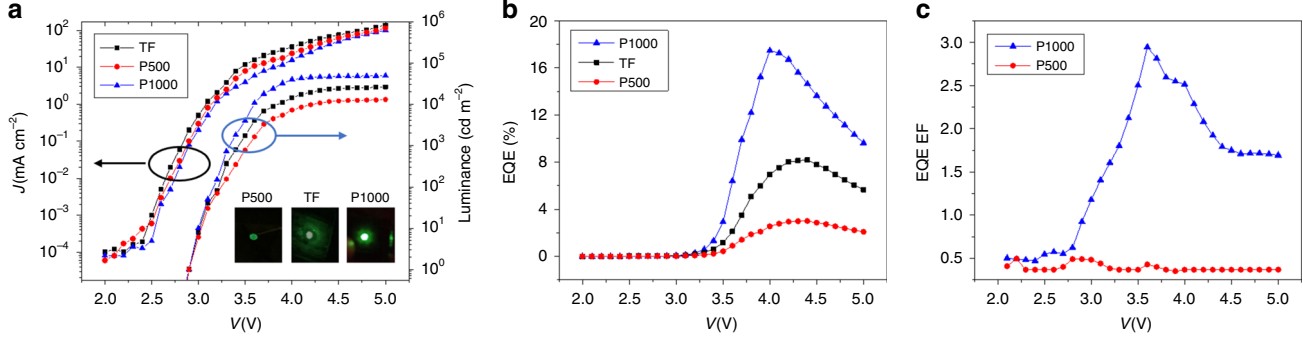

**Fig. 2** Device performance. **a** $J$–$V$ curve, Luminance and **b** External Quantum Efficiency (EQE) of the thin film (TF) and AAM samples. **c** EQE enhancement factor of AAM devices compared with the TF device. Inset figures in **a** are the optical photos for different devices driving at 3.5 V

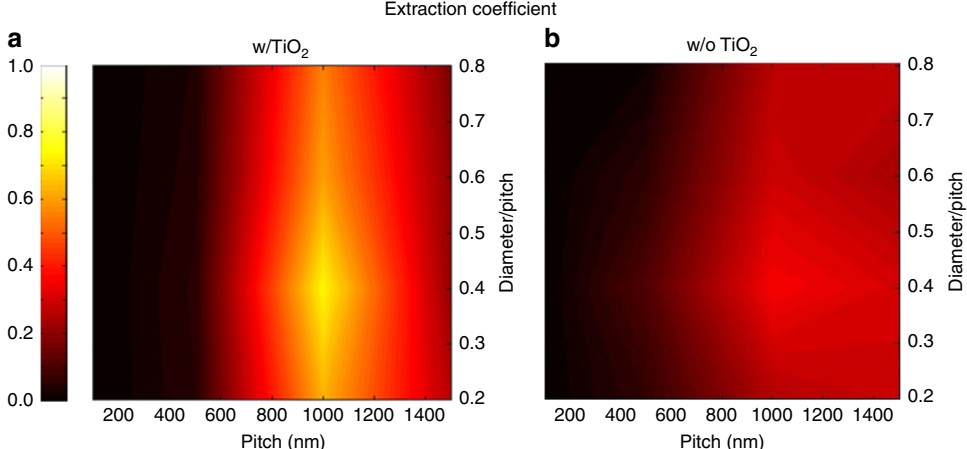

**Fig. 3** Light extraction simulation. Two-dimensional (2D) contour of the extraction coefficient for AAM devices **a** w/ and **b** w/o TiO$_2$. Different pitches and diameter/pitch ratios were simulated. For device w/o TiO$_2$, the AAM channel was filled with air

Figures 5a–e. Figure 5a shows the $E^2$ intensity for the planar structure, and it's easy to see that most of the light energy is confined inside the device. Figure 5b exhibits the $E^2$ intensity for P100 AAM sample, and most of the light is also confined in perovskite active layer as the diameter of nanopores is too small to support any guided modes and thus the light cannot be coupled into TiO$_2$ NWs. And from the mode analysis in Supplementary Figure 10, no guided modes can be formed in P100 AAM. Figure 5c shows the $E^2$ intensity for P500 AAM sample, apparently guided modes are formed inside the TiO$_2$ NWs. From mode analysis shown in Supplementary Figure 11 and Supplementary Note 3, 3 guided modes can be formed in the P500 TiO$_2$ NWs. However, a clear standing wave can be noted in Figure 5c, which means the light energy is confined in the TiO$_2$ NWs. Figure 5d displays the $E^2$ intensity for P1000 AAM, the guided modes were converted to leaky mode which can be extracted out as radiative propagation[30–33]. Figure 5e shows the $E^2$ intensity for P1500 AAM, in which the TiO$_2$ NW diameter is further increased, and it can support more guided modes. As shown in Supplementary Figure 12 and 13, 8 and 12 guided modes can be formed in P1000 and P1500 AAM. Meanwhile, far-field $E^2$ intensities are illustrated in Figures 5f–j. Note that the color bar range in Figure 5i and Figure 5j is one order of magnitude higher than that of Figures 5f–h. It can be clearly seen that the P1000 sample has the largest $E^2$ intensity. Importantly, since only the guided modes interacting with the nanophotonic structure can be extracted out, the optimized diameter is required to support only a proper amount of guided modes.

Figure 6 shows the near field and extinction (defined as 1-transmission) spectra of P500, P1000, and P1500 photonic crystals without NDs. More details can be found in Supplementary Note 5. Figure 6a shows that the energy is confined inside TiO$_2$ NW. And the interference pattern in Figure 6d also confirms the existence of standing wave in the P500 photonic crystals. This is because the light wavelength is twice larger than the TiO$_2$ NW diameter, therefore, the light scattering is very weak. Figure 6b shows a clear pattern of scattering resonance in the near field of P1000 photonic crystal. Moreover, the extinction spectrum of Figure 6e shows a passband centering at the emission peak (530 nm) of our LED device, which means the P1000 photonic crystal is working as optical antennas to extract out the light coupled into the TiO$_2$ NW by NDs coupler. Figure 6c also shows confined energy in the TiO$_2$ NW of P1500 photonic crystal. And from the extinction spectrum shown in Figure 6f, the emission peak is at the edge of the passband of P1500 photonic crystal optical antennas. That is why the P1500 photonic crystal is not efficient in extracting out the light. Moreover, the extinction spectra of P1000 and P1500 photonic crystals optical antennas show different passband positions with respect to LED light emission peak position, which means the proper geometry design is required to obtain the best light extraction efficiency for a certain emission wavelength. These analyses can be clearly

**Table 2 Simulation result of the light extraction coefficient (EC) and its enhancement compared with the planar counterpart for different pitches with aspect ratio of 0.4**

| Geometry | Extraction coefficient (EC) | EC Enhancement factor |
|---|---|---|
| Planar | 0.1 | 1 |
| Pitch 100 nm, diameter 40 nm (40 V AAM) | 0 | 0 |
| Pitch 500 nm, diameter 200 nm (200 V AAM) | 0.052 | 0.52 |
| Pitch 1000 nm, diameter 400 nm (400 V AAM) | 0.736 | 7.36 |
| Pitch 1500 nm, diameter 600 nm (600 V AAM) | 0.269 | 2.69 |

**Table 3 Device peak external quantum efficiency (EQE) and photoluminescence quantum efficiency (PLQY) of our three generations devices during the course of material optimization**

| Device | TF EQE | P1000 EQE | EQE enhancement factor | TF PLQY | P1000 PLQY | Material engineering |
|---|---|---|---|---|---|---|
| 1st generation | 0.47% | 2.38% | 5.06 | 5% | 4% | N.A. |
| 2nd generation | 3.72% | 11.16% | 3.00 | 40% | 25% | Crystal pinning |
| 3rd generation | 8.19% | 17.50% | 2.14 | 85% | 35% | Crystal pinning + long-chain additive |

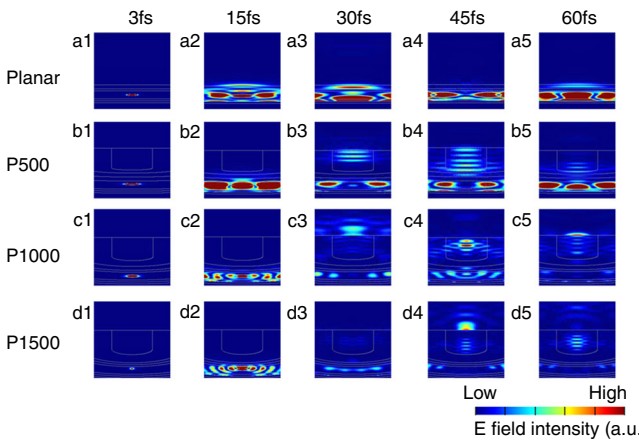

**Fig. 4** Time domain $E$ filed evolution. Light is propagating upwards. Different rows correspond to different structures. From top to bottom: **a1–5** planar, **b1–5** P500 AAM, **c1–5** P1000 AAM and **d1–5** P1500 AAM. Different columns correspond to different moments when the light propagates (from left to right: 3 fs, 15 fs, 30 fs, 45 fs, and 60 fs)

supported by the light propagation animations in the Supplementary Movie 1–5. Additionally, the simulated far-field patterns can also be supported by the far-field measurement experiments (Supplementary Figure 17 and 18 and Supplementary Note 6). Moreover, the angular emission from the P1000 device is also enhanced compared to the thin film and P500 device (Supplementary Figure 19) and related discussion can be found in Supplementary Note 7.

## Discussion

Metal halide hybrid perovskites are promising materials for LED application, however, the conventional planar device structure sets a constraint for light outcoupling. In this work, we have combined nanophotonic structures and thin film perovskite material to fabricate efficient perovskite LED devices. Intriguingly, systematic modeling shows that this nanophotonic substrate is a combination of NDs light coupler and NWs photonic crystal optical antennas. With the help of these two optical

components, the light extraction can be enhanced substantially, and thus we have achieved a record high $CH_3NH_3PbBr_3$ LED EQE of 17.5%, which is twice of that of the planar control device. This significant improvement was systematically investigated with FDTD simulations which confirmed the experimental results. In addition, FDTD simulations also have revealed that the photonic crystal optical antennas convert guided modes (confined energy) to leaky modes (scattering resonance) and leads to the enhanced light outcoupling for the nanostructured device. This study can largely benefit design and fabrication of high-performance perovskite LED devices, and the understanding and insight obtained here can be adopted to other material systems to further enhance the performance of many optoelectronic devices in the future. Meanwhile, the device lifetime is another issue that requires further efforts from all perovskite LED researchers. (The stability tests for our devices at 4 V is shown in Supplementary Figure 20). Besides, the perovskite deposition and material engineering methods that are more compatible with nanostructured substrates can further improve the device performance in the future (Supplementary Note 8). Last but not least, although the light extraction efficiency of 73.6% obtained in this work is already very high, we believe further optimization can be achieved in terms of nanophotonic design which can lead to close to unity light extraction in the future.

## Methods

**Materials**. Poly(3,4-ethylenedioxythiophene): poly(styrenesulfonate) (PEDOT: PSS) pellets, Poly(9,9′-dioctyl-fluorene) (F8), $CH_3NH_3Br$ (MABr), $PbBr_2$, $n$-Butylammonium bromide ($n$-BABr) and dimethylformamide (DMF) were purchased from Sigma Aldrich. $TiO_2$ paste was purchased from Dyesol.

**Device fabrication**. AAM substrates with different geometries were fabricated with a low-cost two-step anodization method reported elsewhere[34–37]. 200 V and 400 V voltages were applied to obtain hexagonal nanopore array with 500 nm and 1000 nm pitches, respectively. Then AAM pores are filled with $TiO_2$ by the following steps: (1) $TiO_2$ paste was dropped onto the AAM chip, (2) the AAM chip was spin coated with 3000 rpm for 50 s, (3) the AAM chip was put in desiccator and pumped for 2 h to remove the air in AAM pores, (4) the AAM chip was wiped with tissue to completely remove the top residue of $TiO_2$ paste. After $TiO_2$ filling, the chip was baked at 110 °C for 20 min and then 500 °C for half hour. Then the AAM chip was flipped over and bonded to glass substrate with UV-epoxy Norland Optical Adhesive (NOA) 81 followed by UV curing for 20 mins. Afterword, aluminum was etched away in saturated $HgCl_2$ solution to expose the AAM ND bottom. ITO was sputtered onto the AAM bottom side with 48 W d.c. power and a gas flow of 50 sccm Ar and 0.5 sccm $O_2$. Then the sample was cleaned by $O_2$ plasma for 2 mins. PEDOT: PSS pellets were dispersed in water to give a

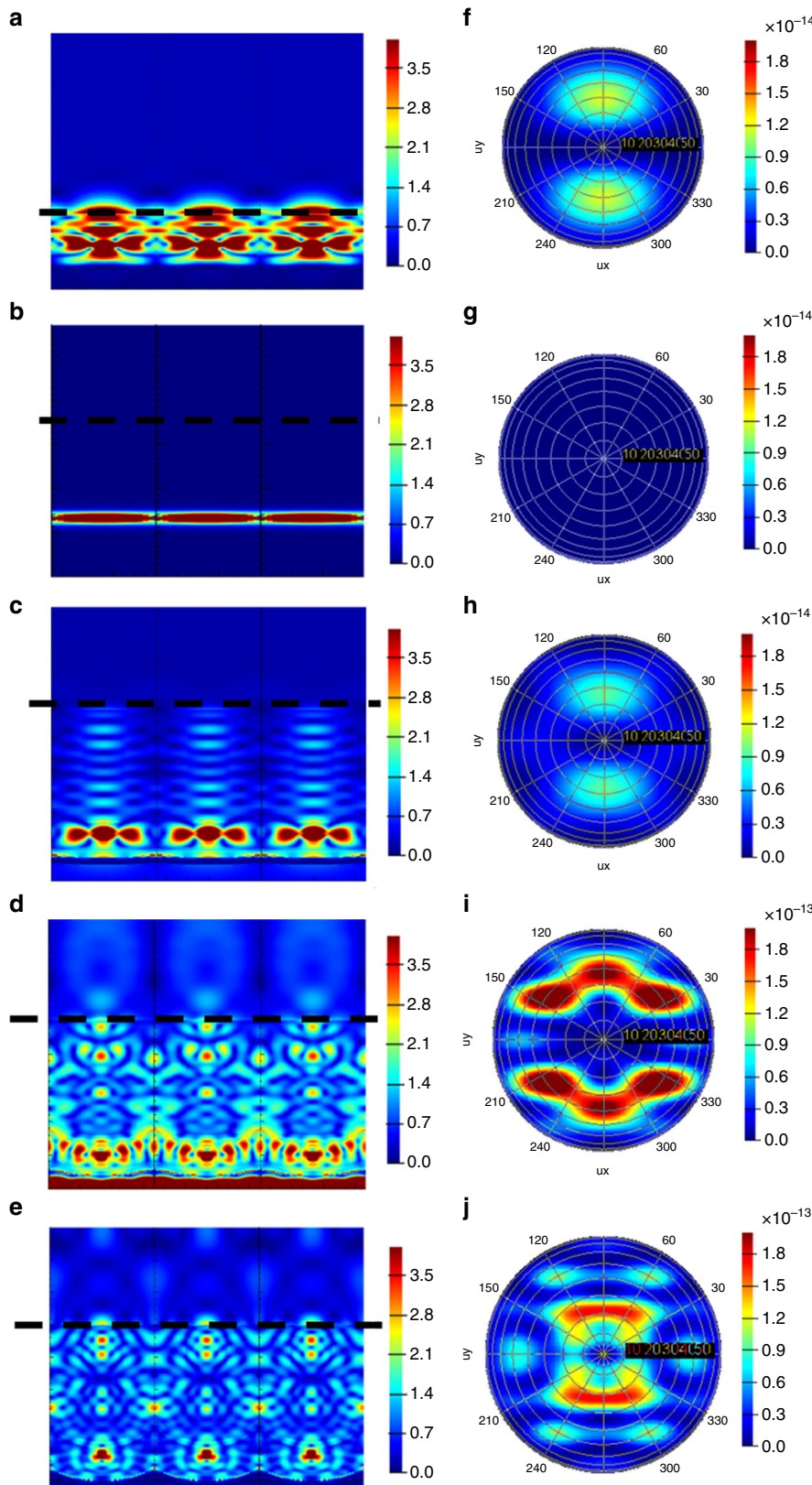

**Fig. 5** Cross-sectional $E^2$ intensities and far-field $E^2$ intensities. Cross-sectional $E^2$ intensity $(V\ m^{-1})^2$ of **a** planar, **b** P100, **c** P500, **d** P1000 and **e** P1500 AAM devices. Dash lines show the light output plane. Simulated far-field $E^2$ intensity of **f** planar, **g** P100, **h** P500, **i** P1000 and **j** P1500 AAM devices. For **f**, **g** and **h** the color bar upper limit is $2 \times 10^{-14}\ (V\ m^{-1})^2$. For **i** and **j**, the color bar maximum is $2 \times 10^{-13}\ (V\ m^{-1})^2$

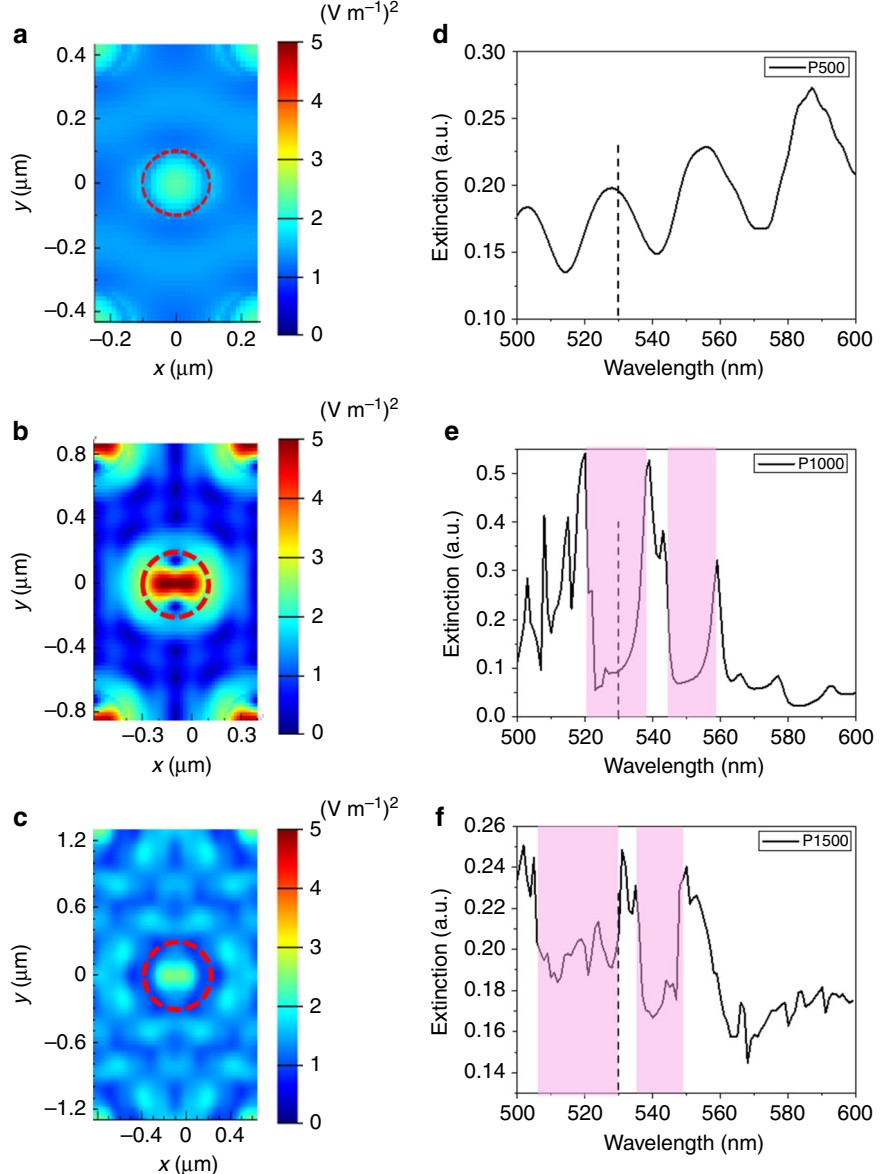

**Fig. 6** Near field and extinction (1- transmission) spectra of the photonic crystal optical antennas. Near field of **a** P500, **b** P1000 and **c** P1500 photonic crystals. Extinction of **d** P500, **e** P1000 and **f** P1500 photonic crystals

concentration of 1.3 wt.%. Then the PEDOT: PSS solution (1.3 wt.% in $H_2O$) was filtered with 0.2 μm pore size filter and then spin-coated onto ITO at 3,000 rpm for 30 s. PEDOT: PSS film was then annealed at 140 °C for 10 min. The Br-Pero solution was made by mixing BABr, MABr, and $PbBr_2$ with molar ratio 0.2:1.05:1 in DMF (500 mg ml$^{-1}$). Then Br-Pero solution was spin coated onto PEDOT: PSS at 3,000 rpm for 50 s. 500 μl toluene was poured onto the sample at 35 s during spin coating of perovskite. Then the sample was baked at 90 °C for 5 mins. F8 was dissolved in chlorobenzene (10 mg ml$^{-1}$), and then F8 solution was spin-coated onto Br-Pero at 3,000 rpm for 30 s. After keeping the sample in dry box for 4 h, Ca (2 nm) and Ag (60 nm) were deposited at 1 Ås$^{-1}$ by thermal evaporation with 3 mm diameter circular mask. In Figure 1 and Supplementary Figure 2, we show three different devices: planar device, P500 AAM device and P1000 AAM device. ITO thickness is around 300 nm, PEDOT: PSS and F8 thicknesses are both around 50 nm, and perovskite thickness is around 400 nm.

**Simulation**. Finite-Difference Time-Domain (FDTD) simulation was carried out with the Lumerical FDTD software package. Periodic boundary conditions were applied to form the hexagonal periodic array. Point light sources inside Br-Pero layer were utilized to simulate the emission. Monitors were added to record the light source power and the far-field light power. Then the extraction efficiency was extrapolated by the ratio of the far-field light power over the light source power. Cross-sectional E field monitors were added to analyze the E field pattern inside the

photonic engineering structures. Time monitors were added to exhibit the light evolution inside the photonic engineering structures. The simulation background was set with refractive index $n = 1.5$ to approximate the UV epoxy. Simulation layout is shown in Supplementary Figure 3.

**Measurement**. SEM was done with JEOL7100F. PL and TRPL were done with Edinburgh Instruments FLS920P. LED performance was measured with a source-measurement-unit Keithley 2400 and a spectrometer (Ocean Optics, FLAME-S-VIS-NIR-ES). EQE was measured by putting the device on the circular side wall window (1 cm diameter) of the integrating sphere (Ocean Optics, FOIS-1). The system was calibrated by the standard light source (Ocean Optics, HL-3P-INT-CAL). The current–voltage (I–V) curves were measured with a probe station equipped with Keithley 4200. X-ray diffraction was done with Bruker D8 X-ray Diffractometer using Cu$K\alpha$ radiation. UV-vis was done by a homebuilt system with Perkin-Elmer 500 spectrometer, an integrating sphere and a broadband halogen light.

## Data availability

All the relevant data are available from the corresponding author upon reasonable request.

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

## Acknowledgements
Q. Zhang thanks Kwong-Hoi Tsui, Jiaqi Chen, Yuefei Cai, and Lian Duan from Department of ECE, HKUST for their technical support and fruitful discussions. Q. Zhang thanks Ms Yan Zhang and Mr Alex H. K. Wong from MCPF, HKUST for their help and discussion during SEM taking. This work was supported by National Natural Science Foundation of China (Project No. 51672231), Shen Zhen Science and Technology Innovation Commission (Project No. JCYJ20170818114107730) and Hong Kong Research Grant Council (General Research Fund Project No. 16237816, 16309018). The authors also acknowledge the support from the State Key Laboratory on Advanced Displays and Optoelectronics and the Center for 1D/2D Quantum Materials at HKUST.

## Author contributions
Z. Fan and Q. Zhang conceived and designed the experiments. Q. Zhang and M.M. Tavokoli performed the device fabrication and measurements. L. Gu, D. Zhang, Y. Lin, S.-F. Leung and S. Poddar contributed to substrate preparation. L. Tang, Y. Gao and Y. Fu contributed to TiO$_2$ preparation. Q. Zhang and J. Guo performed the simulations. Q. Zhang, L. Gu, Y. Lin, and Z. Fan wrote the manuscript. All authors contributed to result discussions and manuscript revisions.

## Additional information

**Competing interests:** The authors declare no competing interests.

