## [Peer Review File · Nature Communications]

Reviewers' comments:

Reviewer #1 (Remarks to the Author):

In this paper, Zhang et al. experimentally demonstrate high efficiency thin film metal halide perovskite light-emitting diodes using nanophotonic templates. Anodic alumina membranes (AAMs) filled with TiO₂ are adopted as two-dimensional nanophotonic substrates. A high external quantum efficiency (EQE) of 17.5% and extraction efficiency of 73.6% are achieved in their work. It certainly is an effective approach for further increase of the EQE of LEDs. But in consideration of the results recently published, Pero-LEDs with EQEs higher than 20% have been realized [1-3]. It is difficult for me to recommend this work in its current shape for Nature Communications, unless significant improvements have been achieved later on.

1. As demonstrated in the manuscript, with the help of nanophotonic templates, the light extraction efficiency is numerically calculated to be 73.6%. And the EQE of the device was finally experimentally measured to be 17.5%. Then we can simply calculate the internal quantum efficiency equal to $17.5\%/73.6\%=23.78\%$. As far as I know, internal quantum efficiencies for pero-LEDs on records are much higher than this, and a nearly 100% of internal quantum efficiency was demonstrated very recently [3]. So, there is still great potential for the authors to improve the performance of their devices.

2. Nanodome shaped alumina barrier layer is adopted in this work to help couple the light from the active material to the AAM channels. It is a good strategy for efficient out coupling of waveguide mode light in the active material, but the geometry parameters like height and radius have not been optimized and more theoretical analyses are needed. The authors should do more analyses and test, which may greatly further improve the performance of their devices. From the viewpoint of optical physics, this method has been well established and understood. So the novelty herein may be greatly discounted.

3. It will be better to have a more concrete study about the guide mode in TiO₂ channels with mode analyses. Better understanding and optimization will be achieved from that, which will help a lot for the design of the photonic devices.

[1] Lin, K. et al. Perovskite light-emitting diodes with external quantum efficiency exceeding 20 percent. Nature 562, 245–248 (2018).

[2] Cao, Y. et al. Perovskite light-emitting diodes based on spontaneously formed submicrometre-scale structures. Nature 562, 249–253 (2018).

[3] Zhao, B. et al. High-efficiency perovskite–polymer bulk heterostructure light-emitting diodes. Nature Photonics (2018).

Reviewer #2 (Remarks to the Author):

In this work, the authors employed a nanodome structured anodic alumina membranes (AAMs) to improve the light extraction of CH₃NH₃PbBr₃ (Br-Perov) based LED. The authors claim to be first to employ nanophotonic substrate in Br-Perov LED and systematically investigated the influence of nanodome pitch size on the performance of Br-Perov LED. The optimum device with a pitch size of 1000 nm (P1000) exhibited an unprecedentedly high EQE of 17.5%. The simulation results are highly convincing and generally agrees well with the experimental outcomes, revealing a 75.6% improvement in the light extraction efficiency of nanodome AAMs based Pero LED device. However, the LED characterization part in the manuscript severely lacks of experimental evidence and scientific elucidation to fully support the performance improvement of Pero LED. There are also other issues and questions which require the further attention of the authors. Major revisions are to be conducted on the manuscript before acceptance can be considered:

1. In Page 3. "...overall EQE of Br-Pero LEDs is still low in .." It is advisable to provide a numerical range/exact figure for a statement like this.
2. Figure 1 and Figure 2 are showing almost similar, redundant morphology information of nanodome AAMs based Br-Pero LED.
3. In Figure 3, the enhancement factor (EF) of EQE has already been listed in Table 2. Figure 3(d) can be omitted, it is redundant and does not carry any significance. Also, the authors should consider to merge Figure 3(a) and (b) for better comparison on the J-V-L characteristics of Pero LED.
4. The author needs discussion for the optical and electrical properties of the AAM based ITO electrode and comparison reference ITO TF.
5. In overall, the device characterization data (only EQE improvement) for Pero LED is insufficient to substantiate its light extraction improvement. The authors should also include the data especially 'Total EQE' measured by using integrating sphere or "angle dependent light emission" of Pero LED as these are the two basic evaluation parameters for light extracted LED device.
6. No explanations/mechanisms were provided on how the geometry and aspect ratio of nanodome AAMs affect the light extraction and luminance of Pero-LED. Why is there such a drastic difference in the device performance of P1000 and P500 nanodome MMAs?
7. Typically, the use of a nanophotonic substrate could also help to enhance the angular emission of an LED device. Is there any information on the angular EL emission changes of Br-Pero LED employing AAMs nanodome? (related reference: Adv. Mater. 25, 3571-3577, 2013; Org. Electron. 57, 201-205, 2018)
8. Despite the high EQE value at 4.0V, the increasing/declining trend of EQE exhibits a drastic roll-off particularly for P1000 device. The authors should provide explanation for this phenomenon.
9. Judging from the EQE curve, the optimum P1000 Pero-LED is seemingly suffering from short device lifetime. Please comment on this.
10. To enable readers to perceive the current progress of Pero LED, it is advisable to list the performance (EQE) of representative Pero LEDs (of almost similar material system) in a table.
11. There are numerous technical errors in the manuscript:
 - The denotation of 'Ca/Au' in the caption of Figure 1(a).
 - The label of 'MAPbBr₃' in Figure 1(c).
 - Figure 3 (a) should be denoted as 'J-V' curve.
 - The denotation of 'Br-Pero' should come first in Page 3, Line

 - No information on the denotation of 'TF' in Figure 3.
 - In Page 7, Figure 4d mismatch with manuscript
 - In Figure 3, each sample needs same color, i.e. TF: black line, P500: red line and P1000: blue line.
12. More related recently published references regarding nanophotonic structures/perovskite LED should be added in the manuscript: Journal of Materials Chemistry C 6 (20), 5444-5452, 2018; Nanoscale 10 (34), 16184-16192, 2018.

Reviewer #3 (Remarks to the Author):

Zhang et al build nanophotonic devices to enhance outcoupling from perovskite solar cells. They demonstrate that they could hypothetically gain a 7x improvement based on their modeling, and observe a 2x improvement in practice. I find the modeling quite interesting and well done, but the device side and the connections between the two are lacking. Therefore, I believe significant revisions are required before publication could be considered.

First, the device work is incomplete. The authors need to provide device statistics – how many were made, and what was the spread in efficiencies? This is particularly important as their nanophotonics improvement hinges on the EQE improvement. Next, the authors need to provide more information on how they measured the EQE. Finally, the authors should provide stability data. It's not the focus of the paper, but it's an important piece of modern LEDs and should be included in the SI.

Second, the connection between the devices and the models is lacking. The authors have wonderful far-field simulations of the emission – this should be straightforward to measure and show if the fabricated devices match. The authors need to do so. Further, there's a large disagreement between the model and devices. The authors cite 10% maximum for planar devices, yet many groups report 15+% EQE, and even the 8% reported by the authors indicate this value is too low. Then the authors claim there should be a 7x increase in outcoupling, but they only see a factor of two. Where are the extra losses in real devices vs the model coming from?

Significant editing issues, including:

- A) The authors mean LUMO, not LOMO pg 5.
- B) Please label the colors in Figure 2.
- C) The authors switch color schemes in different panels of Figure 3, making it hard to compare devices.
- D) The simulations are very nice (fig S6/videos). It would be nice if the authors included the timestamp of the excitation (rather than the simulation). I.e, 3 fs, 6 fs, etc.

Reviewers' comments:

Reviewer #1 (Remarks to the Author):

*In this paper, Zhang et al. experimentally demonstrate high efficiency thin film metal halide perovskite light-emitting diodes using nanophotonic templates. Anodic alumina membranes (AAMs) filled with TiO₂ are adopted as two-dimensional nanophotonic substrates. A high external quantum efficiency (EQE) of 17.5% and extraction efficiency of 73.6% are achieved in their work. It certainly is an effective approach for further increase of the EQE of LEDs. But in consideration of the results recently published, Pero-LEDs with EQEs higher than 20% have been realized (Lin, K. et al. Perovskite light-emitting diodes with external quantum efficiency exceeding 20 percent. *Nature* 562, 245–248 (2018); Cao, Y. et al. Perovskite light-emitting diodes based on spontaneously formed submicrometre-scale structures. *Nature* 562, 249–253 (2018); Zhao, B. et al. High-efficiency perovskite–polymer bulk heterostructure light-emitting diodes. *Nature Photonics* (2018)). It is difficult for me to recommend this work in its current shape for *Nature Communications*, unless significant improvements have been achieved later on.*

Response:

We deeply appreciate the reviewer for the encouraging comment and agreeing on the effectiveness of nanophotonic approach on improving EQE of LEDs. However, we want to emphasize again that in this work we used MAPbBr₃ material system as a model material to demonstrate the effectiveness of our nanophotonic structure and we have been able to improve the EQE record of this material system substantially from 9.3% to 17.5%. On the other hand, current 20% EQE record is from a different material system, namely Cs/FA perovskite. Therefore, direct comparison between our material system with the Cs/FA perovskite material is NOT perfectly fair. As a matter of fact, our nanophotonic approach is a generic approach, combining with higher efficiency materials such as Cs/FA perovskite can certainly lead to record breaking device performance, i.e. EQE, this is the true significance of our work. And this is what we will work on in the future.

1. As demonstrated in the manuscript, with the help of nanophotonic templates, the light extraction efficiency is numerically calculated to be 73.6%. And the EQE of the device was finally experimentally measured to be 17.5%. Then we can simply calculate the internal quantum efficiency equal to $17.5\%/73.6\%=23.78\%$. As far as I know, internal quantum efficiencies for pero-LEDs on records are much higher than this, and a nearly 100% of internal quantum efficiency was demonstrated very recently [3]. So, there is still great potential for the authors to improve the performance of their devices.

Response:

The reviewer is definitely correct that PLQY of the perovskite thin film on nanostructured substrate is not very high and we do have some potential for improvement in the future. (PLQY of our device is shown in Table S2.) As explained in Table S2, we have gone through three generations of optimization processes following literature reports to improve planar thin film PLQY from 5% all the way to 85% which is relatively high already. However, the optimal planar thin film process conditions do not necessarily yield the thin films with ideal quality on nanostructures due to nanostructured surface morphology. Therefore, the PLQY of perovskite on nanostructures was only improved from 4% up to 35%. This reminds us that perovskite process conditions on nanostructures need further optimization which has not been reported

before. For example, vapor deposition method that can give uniform and conformal material coating might be a solution to accommodate the nanostructured substrates. And this also means there is still great potential for us to improve the performance of our devices beyond current record, as the reviewer has pointed out.

2. Nanodome shaped alumina barrier layer is adopted in this work to help couple the light from the active material to the AAM channels. It is a good strategy for efficient out coupling of waveguide mode light in the active material, but the geometry parameters like height and radius have not been optimized and more theoretical analyses are needed. The authors should do more analyses and test, which may greatly further improve the performance of their devices. From the viewpoint of optical physics, this method has been well established and understood. So the novelty herein may be greatly discounted.

Response:

We thank the reviewer for the valuable suggestion. Actually, we have provided the geometry effect data in Figure 3. The AAM dome shape is determined by the AAM pitch, as shown in Figure S7, the pitch p is the x-axis in Figure 3, and the DP ratio (Diameter/pitch = $\frac{2r_1}{p}$) is y-axis in Figure 3. As the nanodome diameter r_2 is half of the pitch p , when we tune the pitch of AAM, diameter of nanodome changes accordingly. Simply speaking, we have changed pitches and diameters of the AAM pores in the simulation part.

As for the experiment part, we measured the performance of devices with different pore sizes from 300 nm to 600 nm for P1000 AAM device. The expanded data is provided as Figure S9. We studied the geometry effect of AAM substrate, and changed not only the pitches, but also the channel diameters. Because P1000 AAM gave us the best EQE, we chose the P1000 AAM for diameter effect study. DP ratio of 0.3, 0.4, 0.5 and 0.6 corresponds to AAM pore size of 300 nm, 400 nm, 500 nm and 600 nm, respectively. And we found the peak EQE for DP0.3, DP0.5 and DP0.6 are 0.96 times, 0.94 times and 0.82 times of that of the optimized DP0.4, which is quite consistent with our simulation result shown in Figure 3a.

Regarding the novelty of our nanophotonic approach, the combination of nanodomains with photonic crystals has **NOT** been reported to our best knowledge. As for the optical physics in terms of the nanodomains, most nanodome structures used for LEDs are working as back scatter or surface scatter.[1-4] Among them, Yamada's work and Tadatomo's work are very classical works using nanostructures as back scatter for GaN LED. However, the nanodomains in our structure is an optical mode coupler component, working together with photonic crystal optical antennas, which is quite unique.

To support this, we have added new simulation results of light extraction with only nanodomains and show results in Figure S16. The focusing point created by nanodomains can be clearly resolved in Figure S16 (g)-(i). With the nanodomains only, we can obtain 20%, 46% and 43.5% light extraction for P500, P1000 and P1500 AAM devices, respectively. All of them are already better than thin film reference devices. However, if the photonic crystal structure is combined with the nanodomains, a much higher light extraction of 73.6% can be achieved. Intriguingly, this enhancement from photonic crystal only happens when the geometry is properly designed. As can be seen from Table 2, the light extraction for P500, P1000 and P1500 devices combining both nanodomains and photonic crystal are 5.2%, 73.6% and 26.9%, respectively. This is a very strong evidence that the P1000 photonic crystal is superior in extracting the guided modes than P500 and P1500 AAMs.

To give more details of this point, we also simulate the near field and light extinction of the photonic crystals only and show results in Figure S17. Figure S17a-c shows the near field of the P500, P1000 and P1500 photonic crystals, respectively. For P500 and P1500, it can be seen that the light energy is confined inside the TiO₂ channels. The field pattern of P1000 photonic crystal near field (Figure S17b) shows a scattering resonance (leaky mode). [5] The scattering resonance is also an evidence of the optical antennas effect which can effectively convert the confined light energy to propagation radiation, and the antennas effect can also be supported by the extinction spectrum pass band centering at 530 nm (EL peak).[6, 7] As for the P500 photonic crystal, the extinction (Figure S17d) shows a typical interference pattern, which indicates the standing wave (guided modes) is confined inside TiO₂ NWs, which can also be supported by Figure S16a and Figure S17a. And if we look at the extinction of P1500 (Figure S21f), 530 nm is at the edge of a pass band, which means this wavelength is not in the working range of P1500 optical antennas, and that is why most of the light energy is confined in the TiO₂ NWs, as shown in Figure S17c. Basically, this explains why adding photonic crystals can help the light extraction of P1000 AAM device but not for P500 and P1500 AAM device.

As the reviewer mentioned, the AAM nanodomains mainly work as focusing lens to couple light from active material to the AAM channels. More importantly, whether the generated guided modes in AAM channels can be extracted out really depends on the geometry of AAM pitch and pore diameter. With our mode analysis (Figure S10-13) in supporting information, our hypothesis can be strongly supported. The P100 AAM supports no guided modes, therefore light cannot be coupled into it. When AAM pitch increases, P500 AAM starts to support some guided modes. But the scattering effect is too weak (because the TiO₂ NW diameter is two times smaller than wavelength) in P500 AAM, so the guided modes cannot be effectively extracted out. Then the AAM pitch further increases, scattering effect becomes stronger in P1000 AAM, so the guided modes can be converted to leaky modes and then be extracted out, which gives the optimized light extraction efficiency. However, the pitch cannot be infinitely increased because larger pitch AAM means more guided modes. If the guided modes cannot be extracted out by scattering, they will finally be dissipated at metal contacts or during non-radiative recombination process. Another very important thing is that our photonic crystal structures are isolated from the active material. The reason is that if the photonic crystal is formed in the LED active layer, the uniformity of the active area will be significantly compromised, which will cause inhomogeneous current distribution undermining the device performance[8]. Importantly, by using the AAM substrate, the active area loss problem can be avoided because the whole device structure was fabricated on top of the photonic engineering substrate rather than inside of it.

Overall, the reviewer is correct that the nanodomains have been reported even though not for Perovskite LEDs. However, the combination of nanodomains light coupler together with photonic crystals optical antennas has never been reported to our best knowledge. Furthermore, this combination of two optical components is formed by low-cost and highly controllable process. More importantly, we provided deep insight into the light extraction mechanisms for nanophotonic substrates with different geometries. That helps other researchers to improve the perovskite LED performance using a brand new and effective strategy.

3. It will be better to have a more concrete study about the guide mode in TiO₂ channels with mode analyses. Better understanding and optimization will be achieved from that, which will help a lot for the design of the photonic devices.

Response:

We sincerely thank the reviewer for the great suggestion. Therefore, we have done systematic mode analysis and added the results as Figure S10-13. And we also added the following discussion to our supporting information.

From Figure S10- Figure S13, we show the modes in different geometry AAMs. Note that here we only discuss the guided modes in the TiO_2 core. And only the first 20 modes (from fundamental modes to high order modes) are calculated.

As for P100 AAM (Figure S10), there is no fundamental TE or TM mode. And the only guided mode 3 has an imaginary effective index and suffers from very high loss. This phenomenon demonstrates our point that the P100 AAM support no guided modes, and therefore the light extraction for P100 AAM is very poor.

As for the P500 AAM (Figure S11), there are 3 guided modes (mode 1, 2 and 9) in the center TiO_2 core. Mode 1 is TM_0 fundamental mode, mode 2 is TE_0 fundamental mode, and mode 9 is hybrid mode.

As for the P1000 AAM (Figure S12), there are 8 guided modes (mode 1, 2, 5, 10, 11, 14, 15, 16). Mode 1 is TE_0 fundamental mode, and mode 2 is TM_0 fundamental mode. Other modes are hybrid modes. Mode 17, 18, 19 and 20 can be classified as leaky modes, as discussed by our previous work.[9] These leaky modes are favorable for light to be extracted out and become radiation propagation. Mode 17 is TM_1 mode, and mode 18 is TE_1 mode.

As for the P1500 AAM (Figure S13), there are 12 guided modes (mode 1, 2, 7, 12, 13, 14, 15, 16, 17, 18, 19 and 20). Mode 1 is TE_0 fundamental mode, and mode 2 is TM_0 fundamental mode. Mode 19 is TE_1 mode, and mode 20 is TM_1 mode. Other modes are hybrid modes.

The mode analysis for P1000 and P1500 AAM also supports our point in the main text that the P1500 AAM supports more guided modes than P1000 AAM. As only the guided modes scattered by/ interacted with the nanostructures can be extracted out, therefore a proper amount of guided modes is required. This also supports the result that P1500 AAM is not as good as P1000 AAM in terms of light extraction.

Reviewer #2 (Remarks to the Author):

In this work, the authors employed a nanodome structured anodic alumina membranes (AAMs) to improve the light extraction of CH₃NH₃PbBr₃ (Br-Perov) based LED. The authors claim to be first to employ nanophotonic substrate in Br-Perov LED and systematically investigated the influence of nanodome pitch size on the performance of Br-Perov LED. The optimum device with a pitch size of 1000 nm (P1000) exhibited an unprecedentedly high EQE of 17.5%. The simulation results are highly convincing and generally agrees well with the experimental outcomes, revealing a 75.6% improvement in the light extraction efficiency of nanodome AAMs based Perov LED device. However, the LED characterization part in the manuscript severely lacks experimental evidence and scientific elucidation to fully support the performance improvement of Perov LED. There are also other issues and questions which require the further attention of the authors. Major revisions are to be conducted on the manuscript before acceptance can be considered:

Response:

We sincerely thank the reviewer for the highly positive comments and we have performed more experiments and simulations to address the questions raised, below please find our responses.

1. In Page 3. "...overall EQE of Br-Perov LEDs is still low in .." It is advisable to provide a numerical range/exact figure for a statement like this.

Response:

Thanks for the suggestion. Table S1 is added to Supporting Information to summarize the EQEs and light emitting materials of the state-of-art perovskite LEDs. Meanwhile, we also provided their PLQY and reported/estimated light extraction efficiencies.

2. Figure 1 and Figure 2 are showing almost similar, redundant morphology information of nanodome AAMs based Br-Perov LED.

Response:

We thank the reviewer for the suggestion, thus we have rearranged Figure 1 and Figure 2, and move some part of Figure 1 and 2 to supporting information.

3. In Figure 3, the enhancement factor (EF) of EQE has already been listed in Table 2. Figure 3(d) can be omitted, it is redundant and does not carry any significance. Also, the authors should consider to merge Figure 3(a) and (b) for better comparison on the J-V-L characteristics of Perov LED.

Response:

We thank the reviewer for the suggestion and below are the changes we made to the figures:

1) Figure 3a and 3b are merged.

2) Actually, Figure 3d is the enhancement factor of the device EQE, and Table 2 is the enhancement factor of simulated light extraction efficiency. Since they are different data, it might be proper to keep them that way.

4. The author need discussion for the optical and electrical properties of the AAM based ITO electrode and comparison reference ITO TF.

Response:

We thank the reviewer for the excellent suggestion. We have measured the transmittance and sheet resistance of our sputtered ITO on AAM substrate and the reference ITO glass with the same ITO thickness. The transmittance spectrum is added as Figure S4 in SI. It can be seen that the transmittance of our sputtered ITO on AAM substrate is ~ 0.5 at 550 nm wavelength but the planar ITO glass shows 0.8 transmittance. This is a combinational effect of our non-ideal sputtering process and larger actual ITO area on rough nanostructured surface. In this case, we need to sacrifice the transmittance a little bit in order to achieve a good sheet resistance. (The O_2 flow rate is low, less than 1 sccm, during sputtering process.) And we also measured the sheet resistance of our sputtered ITO on AAM substrate and the value is $\sim 90 \Omega/\square$, while the commercial ITO glass has a sheet resistance $\sim 50 \Omega/\square$. And the lower transmittance of our sputtered ITO (0.5 compared to 0.8) also partially explains why our experiment has not achieved as high enhancement factor as the modeling.

5. In overall, the device characterization data (only EQE improvement) for Pero LED is insufficient to substantiate its light extraction improvement. The authors should also include the data especially 'Total EQE' measured by using integrating sphere or "angle dependent light emission" of Pero LED as these are the two basic evaluation parameters for light extracted LED device.

Response:

We thank the reviewer for the valuable suggestion. In fact, our EQE was already measured by using integrating sphere, so they are essentially the total EQE already. Meanwhile, we followed the reviewer's suggestion and performed angular emission, and the results are shown in Figure S20. From the angular emission spectra, we can tell that our devices, no matter thin film or nanostructured ones, basically follow the trend of the Lambertian profile (Figure S20b). Generally, the overall EL intensity of P1000 AAM device is two times of that of the TF one. Note that the nanostructured devices show a relatively reduced EL intensity at emission angle larger than 30° , which is more obvious in Figure S20a. This can be understood because the large angle emission is focused by the AAM dome shapes which can be regarded as focusing lens, which can be supported by our light propagation videos in supplementary. The AAM nanodomains help to couple the light from perovskite layer into the AAM channels (TiO_2) and form guided modes. Basically, these guided modes propagate vertically. Thanks to the photonic crystal optical antennas effect, the guided modes can be converted to leaky modes to be extracted out by a proper geometry (P1000), therefore, the light will emit in random directions again after being scattered by the nanophotonic substrates.

The angular emission spectrum does not show significantly modified patterns for our AAM based LEDs. One possible reason is that for our case the photonic crystal structures are not inside the active materials like most photonic crystal LED. Actually, this can be an advantage that the emission pattern is close to Lambertian profile.

6. No explanations/mechanisms were provided on how the geometry and aspect ratio of nanodome AAMs affect the light extraction and luminance of Pero-LED. Why is there such a drastic difference in the device performance of P1000 and P500 nanodome MMAs?

Response:

We thank the reviewer for the question. Actually, we have provided the geometry effect data in Figure 3. The AAM dome shape is determined by the AAM pitch, as shown in Figure S7, the pitch p is the x-axis in Figure 3, and the DP ratio (Diameter/pitch = $\frac{2r_1}{p}$) is y-axis in Figure 3. As the nanodome diameter r_2 is half of the pitch p , when we tune the pitch of AAM, diameter of

nanodome changes accordingly. Simply speaking, we have changed pitches and diameters of the AAM pores in the simulation part. However, as the AAM dome shape is only determined by the pitch, so it might not be easy to change the aspect ratio of the nanodomains independently in our case.

The significant light extraction difference of P1000 and P500 AAM devices come from the different nanophotonic processes. As we discussed in the maintext, the P500 AAM does not have as strong scattering effect as the P1000 AAM device because the TiO_2 channel diameter of P500 is much smaller than wavelength. In P500 device, the light coupled into TiO_2 channels become standing wave (standing wave pattern can be seen in Figure 4c and Figure S17a. And Figure S17d shows an interference pattern for the P500 photonic crystal which also indicates the standing wave). This point can also be supported by the supplementary videos. When we increased the pitch from 100 nm to 500 nm, 1000 nm and 1500nm, we can see there is no guided mode in the 100nm pitch AAM due to the too small TiO_2 channel diameter, which can be supported by the added mode analysis in Figure S10. Then the 500 nm pitch start to form some guided modes, therefore light can be coupled from perovskite layer into the AAM channels, however, at this pitch, the scattering effect of the photonic crystal is too weak because the feature size (TiO_2 channel diameter) is much smaller than wavelength. As a result, the guided modes (confined energy) will be dissipated in the end (by metal contacts or non-radiative recombination in active layer). Then for 1000 nm pitch AAM, scattering effect from the photonic crystal optical antennas starts to play a key role, and those guided modes can be extracted out when they are converted to leaky mode (scattering resonance, as can be supported by the near field shown in Figure S17b). This can be also supported by Figure 4d and the supplementary video. However, when the pitch is further increased to 1500 nm, more guided modes will be generated, as can be supported by Figure S12 and Figure S13. P1000 AAM has 8 guided modes among the 20 lowest order modes, and P1500 AAM has 12 guided modes. We also discuss this tradeoff between guided modes formation and scattering effect of the photonic crystal optical antennas. Basically, only when guided modes can be formed, the light can be coupled from perovskite layer into AAM channels by nanodomains light coupler. But in order to extract the light out, guided modes must be converted to leaky mode which is radiative. P500 AAM has too weak scattering effect, and P1500 AAM has too many guided modes. Moreover, if we look at the extinction spectrum for P500, P1000 and P1500 shown in Figure S17, the P500 shows no optical antennas effect in the 500-600nm range, P1000 shows a pass band centering at 530nm (emission peak of our device). And for P1500, the emission peak is at the edge of a pass band, which makes the scattering effect of the P1500 photonic crystal optical antennas less effective than P1000. Comparing all the three different AAM geometries, P1000 AAM turned out to be the best trade-off.

7. Typically, the use of a nanophotonic substrate could also help to enhance the angular emission of an LED device. Is there any information on the angular EL emission changes of Br-Perov LED employing AAMs nanodome? (related reference: Adv. Mater. 25, 3571-3577, 2013; Org. Electron. 57, 201-205, 2018)

Response:

We thank the reviewer for the constructive suggestion. Thus, we have added the angular EL emission as Figure S20. If we use the EL intensity at 0° of thin film device as a reference, we can see an enhanced EL intensity for the P1000 AAM device, as shown in Figure S20a. And if we use the EL intensity at 0° of each device as its own reference, we can tell that all the emission profiles basically follow the Lambertian profile. Note that the P1000 device show some reduced

EL intensity at angles larger than 30° as compared to Lambertian profile. We attribute it to the lens focusing effect of the AAM nanodomains. However, the reduction of large angle emission intensity is not too much, which is because the AAM structures scatter the guided modes and make radiation propagation into random directions again. If we compare the absolute EL intensities as shown in Figure S15a, the P1000 AAM has huge enhancement in terms of the angular emission, especially for -45° to 45° range.

8. Despite the high EQE value at 4.0V, the increasing/declining trend of EQE exhibits a drastic roll-off particularly for P1000 device. The authors should provide explanation for this phenomenon.

Response:

In the maintext, we have a few sentences “The EQE increases from 0-4 V, reaches the maximum value at around 4 V and starts to show roll-off in 4-5 V voltage range. The EQE roll-off can be attributed to Auger and trap-assisted recombination as reported previously [10]. Beyond 5V, the devices start to fail due to Joule heating[11]” that discussed about this roll-off effect. It’s true that we should add more discussions.

This roll-off effect is indeed a major issue for most LEDs. And for perovskite LEDs, this roll-off is very common and is mainly caused by non-radiative Auger recombination. This problem can be addressed by applying multiple quantum wells, as reported by W. Zou *et. al.*[12]

Actually, the roll-off for P1000, P500 and TF are quite similar, all dropped to about half of the peak EQE when voltage is increased to 5V. The reason why P1000 device roll-off looks more obvious is mainly because we plot the EQE curves in linear scale, not logarithm scale.

We want to point out that this roll-off is a common problem for not only our device but also even for the devices with 20% EQE record. For example, this roll-off can also be seen in Figure 2c in K. Lin’s Nature paper published recently. [13]

9. Judging from the EQE curve, the optimum P1000 Pero-LED is seemingly suffering from short device lifetime. Please comment on this.

Response:

Indeed, our device lifetime is not particularly long. The stability test result is added as Figure S21. We measured the device EL intensity for two minutes with 4V driving voltage. The normalized EL for the TF device dropped from 1 to 0.7, the normalized EL for P500 AAM dropped from 0.37 to 0.24, and the normalized EL for P1000 AAM dropped from 2.11 to 1.27 in 2 mins. Note that we are using much higher current density (~50 mA/cm²) and luminance (~10,000 cd/m²) to test the stability than many reported works[13, 14], and all of our devices are not packaged, and the measurements were performed in ambient condition with 75% humidity, these are main reasons account for relatively fast device performance decay.

10. To enable readers to perceive the current progress of Pero LED, it is advisable to list the performance (EQE) of representative Pero LEDs (of almost similar material system) in a table.

Response:

It’s a very good suggestion. Table S1 is added to summarize the performance of the state-of-art perovskite LEDs and their emitting materials. In the same time, we also show their PLQY and reported/estimated light extraction efficiency.

11. There are numerous technical errors in the manuscript: - The denotation of ‘Ca/Au’ in the caption of Figure 1(a).

- The label of 'MAPbBr₃' in Figure 1(c).
- Figure 3 (a) should be denoted as 'J-V' curve.
- The denotation of 'Br-Perov' should come first in Page 3, Line
- No information on the denotation of 'TF' in Figure 3.
- In Page 7, Figure 4d mismatch with manuscript
- In Figure 3, each sample need same color, i.e. TF: black line, P500: red line and P1000: blue line.

Response:

We thank the reviewer for having examined our manuscript and pointed out these errors. We have carefully checked our manuscript and all of these errors have been corrected in revision.

12. More related recently published references regarding nanophotonic structures/perovskite LED should be added in the manuscript: Journal of Materials Chemistry C 6 (20), 5444-5452, 2018; Nanoscale 10 (34), 16184-16192, 2018.

Response:

Thanks to the reviewer, these two works are very enlightening. They are added in page 4 discussion, highlighted with red color font.

Reviewer #3 (Remarks to the Author):

Zhang et al build nanophotonic devices to enhance outcoupling from perovskite LEDs. They demonstrate that they could hypothetically gain a 7x improvement based on their modeling, and observe a 2x improvement in practice. I find the modeling quite interesting and well done, but the device side and the connections between the two are lacking. Therefore, I believe significant revisions are required before publication could be considered.

Response:

We thank the reviewer for the positive comment and below please find our response to the questions raised point by point.

First, the device work is incomplete. The authors need to provide device statistics – how many were made, and what was the spread in efficiencies? This is particularly important as their nanophotonics improvement hinges on the EQE improvement. Next, the authors need to provide more information on how they measured the EQE. Finally, the authors should provide stability data. It's not the focus of the paper, but it's an important piece of modern LEDs and should be included in the SI.

Response:

We thank the reviewer for the suggestion, therefore:

1) The statistics results are added as Figure S5. For each kind of device, we counted ~100 devices and plot the histogram of the EQE distribution. The P1000 AAM device has an average EQE of 15.93% with 0.96% standard deviation. The P500 AAM device has an average EQE of 2.26% with 0.73% standard deviation. And the thin film device has 7.05% average EQE with 0.85% standard deviation.

2) Our EQE is measured with a spectrometer (Ocean Optics Flame, LED measurement software package purchased from the same company Ocean Optics) together with an integrating sphere with opening on the side wall. The spectrometer and software package are characterized by the standard light source (Ocean Optics, HL-3P-INT-CAL). Basically, the EQE of our device is the output light power (collected by integrating sphere) divided by the input electrical power (voltage multiplied by current measured from Keithley 2400). Detailed information is added to the Methods part.

3) Stability test is added to Figure S21. We measured the device EL intensity for two minutes with 4V driving voltage. The normalized EL for the TF device dropped from 1 to 0.7, the normalized EL for P500 AAM dropped from 0.37 to 0.24, and the normalized EL for P1000 AAM dropped from 2.11 to 1.27 in 2 mins. Note that we are using much higher luminance to test the stability than many reported works[13, 14], and all of our devices are not packaged, and the measurements were performed in ambient condition with 75% humidity, these are main reasons account for relatively fast device performance decay.

Second, the connection between the devices and the models is lacking. The authors have wonderful far-field simulations of the emission – this should be straightforward to measure and show if the fabricated devices match. The authors need to do so. Further, there's a large disagreement between the model and devices. The authors cite 10% maximum for planar devices, yet many groups report 15+% EQE, and even the 8% reported by the authors indicate this value is too low. Then the authors claim there should be a 7x increase in outcoupling, but

they only see a factor of two. Where are the extra losses in real devices vs the model coming from?

Response:

We thank the reviewer for raising these critical questions.

1) We have just performed the far field measurement and added the result as Figure S18. In order to approximate the linear polarized light source in the simulation and also in order to excite strong enough far field, we chose a linearly polarized laser to excite far field for thin film and nanophotonic samples. It can be found that the measured far field pattern of the P1000 AAM matches very well with the simulation results. The TF and P500 samples don't have clear far field pattern in both simulation and measurements. Intriguingly, when we rotate the laser, the far field pattern of our P1000 sample also rotates. Considering our LED device generates random and non-polarized light when electrically turned on, it's difficult to see the far field pattern from the electrically pumped device. Similar situation is also discussed in Z. Khokhar *et al.*'s work, far field EL of their photonic crystal LED has no clear diffraction spots because light is uniformly generated all over the LED surface and couples collectively to the quasi-photonic crystal structures.[15]

2) In order to explain the difference of enhancement factor between simulation and experiments, we provided the performance of our 3 generations of devices, and the results are added as Table S2. For our 1st generation device, we used pure MAPbBr₃ without any material engineering process. Both planar version perovskite TF and perovskite on P1000 have PLQYs around 5%, but the enhancement factor of their EQE is 5 times, which is close to the 7 times enhancement from simulation results. For the 2nd generation device, we optimized the perovskite layer by crystal pinning method. Then the planar TF perovskite PLQY was improved to 40% however the perovskite on P1000 substrate was only improved to 25%. In this case, the nanophotonic device EQE is only 3 times of the planar TF device. Furthermore, in the 3rd generation device, we were able to improve PLQY of the planar TF perovskite to 85% however the film on nanophotonic structure was only improved to 35%. In this case, our TF device baseline was enhanced to 8.19% (this is close to 9.3% record of this particular material[11]) and the nanophotonic device EQE was improved by 2 times up to 17.5%. Therefore, the EQE enhancement factor discrepancy is mainly from the different PLQY improvement factor which indicates different material quality improvement progress. And we attribute the reason account for this to different substrate surface morphology for planar substrate and nanophotonic substrate. Namely, the optimal planar TF process conditions not necessarily yield the TFs with ideal quality on nanostructures. Therefore, the PLQY of perovskite on nanostructures was only improved from 4% up to 35%. This reminds us that perovskite process conditions on nanostructures need further optimization which has not been reported before. And with further such optimization, device performance is expected to be improve beyond 20%.

Moreover, the ITO transmittance on AAMs is not as good as our purchased ITO glass. As shown in Figure S4, the ITO glass shows 80% transmittance at 550 nm while the ITO on AAM substrate shows only 50% transmittance. This is because of our nanophotonic structure has large surface area, in order to achieve a good electrical conductance, we reduced the O₂ concentration during ITO sputtering which compromises the transmittance. And this ITO transmittance loss also makes the EQE enhancement of our real device not as high as the modelling result. As a matter of fact, when we compare the planar device with 8.19% EQE and the nanophotonic device with 17.5% EQE, and if we can compensate 2.4 times PLQY loss (85%/35%) and 1.6 times (80%/50%) transmittance loss, the nanophotonic device EQE upper limit will be $17.5\% \times 2.4 \times 1.6 = 67.2\%$, which is about 8 times of that of the planar device. And this will be very close to the modelled

performance enhancement factor. However, we must point out and in reality, due to large surface area on nanophotonic structure, ITO cannot be expected to have both low resistance and high transmittance than a planar ITO on glass. And in nanostructured perovskite film, if more internal scattering occurs, non-radiative recombination will also reduce the material PLQY. Thus, the above calculated 67.2% can only be regarded as an upper limit which may not be easily achieved in reality.

Significant editing issues, including:

A) The authors mean LUMO, not LOMO pg 5.

B) Please label the colors in Figure 2.

C) The authors switch color schemes in different panels of Figure 3, making it hard to compare devices.

D) The simulations are very nice (fig S6/videos). It would be nice if the authors included the timestamp of the excitation (rather than the simulation). I.e, 3 fs, 6 fs, etc.

Response:

We thank the reviewer for having carefully examined our manuscript and picked up typos. And the issues (A)-(D) have been addressed in the revised manuscript.

Reference

- [1] Y. M. Song, G. C. Park, S. J. Jang, J. H. Ha, J. S. Yu, and Y. T. Lee, "Multifunctional light escaping architecture inspired by compound eye surface structures: From understanding to experimental demonstration," *Optics express*, vol. 19, pp. A157-A165, 2011.
- [2] Y.-F. Liu, J. Feng, Y.-F. Zhang, H.-F. Cui, D. Yin, Y.-G. Bi, *et al.*, "Polymer encapsulation of flexible top-emitting organic light-emitting devices with improved light extraction by integrating a microstructure," *Organic Electronics*, vol. 15, pp. 2661-2666, 2014/11/01/ 2014.
- [3] M. Yamada, T. Mitani, Y. Narukawa, S. Shioji, I. Niki, S. Sonobe, *et al.*, "InGaN-based near-ultraviolet and blue-light-emitting diodes with high external quantum efficiency using a patterned sapphire substrate and a mesh electrode," *Japanese Journal of Applied Physics*, vol. 41, p. L1431, 2002.
- [4] K. Tadatomo, H. Okagawa, Y. Ohuchi, T. Tsunekawa, Y. Imada, M. Kato, *et al.*, "High output power InGaN ultraviolet light-emitting diodes fabricated on patterned substrates using metalorganic vapor phase epitaxy," *Japanese Journal of Applied Physics*, vol. 40, p. L583, 2001.
- [5] L. Cao, P. Fan, and M. L. Brongersma, "Optical coupling of deep-subwavelength semiconductor nanowires," *Nano letters*, vol. 11, pp. 1463-1468, 2011.
- [6] J. A. Schuller, T. Taubner, and M. L. Brongersma, "Optical antenna thermal emitters," *Nature Photonics*, vol. 3, p. 658, 2009.
- [7] P. Bharadwaj, B. Deutsch, and L. Novotny, "Optical antennas," *Advances in Optics and Photonics*, vol. 1, pp. 438-483, 2009.
- [8] J. J. Wierer Jr, A. David, and M. M. Megens, "III-nitride photonic-crystal light-emitting diodes with high extraction efficiency," *Nature Photonics*, vol. 3, p. 163, 2009.
- [9] B. Hua, B. Wang, M. Yu, P. W. Leu, and Z. Fan, "Rational geometrical design of multi-diameter nanopillars for efficient light harvesting," *Nano Energy*, vol. 2, pp. 951-957, 2013/09/01/ 2013.
- [10] Z. Chen, C. Zhang, X. F. Jiang, M. Liu, R. Xia, T. Shi, *et al.*, "High - Performance Color - Tunable Perovskite Light Emitting Devices through Structural Modulation from Bulk to Layered Film," *Advanced Materials*, vol. 29, 2017.
- [11] Z. Xiao, R. A. Kerner, L. Zhao, N. L. Tran, K. M. Lee, T.-W. Koh, *et al.*, "Efficient perovskite light-emitting diodes featuring nanometre-sized crystallites," *Nature Photonics*, vol. 11, p. 108, 2017.
- [12] W. Zou, R. Li, S. Zhang, Y. Liu, N. Wang, Y. Cao, *et al.*, "Minimising efficiency roll-off in high-brightness perovskite light-emitting diodes," *Nature Communications*, vol. 9, p. 608, 2018/02/09 2018.
- [13] K. Lin, J. Xing, L. N. Quan, F. P. G. de Arquer, X. Gong, J. Lu, *et al.*, "Perovskite light-emitting diodes with external quantum efficiency exceeding 20 per cent," *Nature*, vol. 562, pp. 245-248, 2018/10/01 2018.
- [14] N. Wang, L. Cheng, R. Ge, S. Zhang, Y. Miao, W. Zou, *et al.*, "Perovskite light-emitting diodes based on solution-processed self-organized multiple quantum wells," *Nature Photonics*, vol. 10, p. 699, 2016.
- [15] A. Z. Khokhar, K. Parsons, G. Hubbard, I. M. Watson, F. Rahman, D. S. Macintyre, *et al.*, "Emission characteristics of photonic crystal light-emitting diodes," *Applied optics*, vol. 50, pp. 3233-3239, 2011.

REVIEWERS' COMMENTS:

Reviewer #1 (Remarks to the Author):

the authors have made some clarifications based on my previously raised comments regarding the novelty of optical design and competitiveness in the specification of the performance compared to the state-of-the-art references. that looks helpful, however I am not yet quite convinced by the timeliness and necessity of the publication in the presence of the current know-hows and benchmarks. Nevertheless, I become neutral now.

Reviewer #2 (Remarks to the Author):

This manuscript is a revised version of a previously reviewed manuscript. Authors have now carried out additional experiments to clearly explain. Along with the other revisions included in the text, the manuscript is now ready for publication.

Reviewer #3 (Remarks to the Author):

The authors have answered my concerns appropriately, and I commend them on a very interesting manuscript. There are a few lingering typos (particularly 'filed' instead of 'field' in Fig S15) but I recommend acceptance upon their resolution.